# Distribution of ESBL-producing and carbapenem-resistant *E. coli* and *Salmonella* spp. in retail chicken meat and live bird market sewage in Bangladesh

**Mst. Sonia Parvin**, **Sudipta Talukder**, **Sayra Tasnin Sharmy**, **Md. Mehedi Hasan**, **Md. Taohidul Islam***

Population Medicine and AMR Laboratory, Department of Medicine, Faculty of Veterinary Science, Bangladesh Agricultural University, Mymensingh, Bangladesh

* taohid@bau.edu.bd

## Abstract

A cross-sectional live bird market (LBM) survey was conducted to determine the prevalence and distribution of extended-spectrum β-lactamase (ESBL)-producing and carbapenem-resistant (CR) *E. coli* and *Salmonella* spp. in retail chicken meat and LBM sewage in Bangladesh. *E. coli* and *Salmonella* spp. were identified using culture-based and molecular methods. Isolates were tested for CR by a disk diffusion test; a confirmatory ESBL screening was performed by double disk synergy test. The isolates were screened for ESBL and CR genes using PCR. Prevalence of ESBL-*E. coli* and *Salmonella* spp. in retail chicken meat was 70% and 43.4%, respectively while in LBM sewage, it was 79.7% and 28.1%, respectively. Carbapenem resistance was also common, detected in 54.1% and 46.9% of *E. coli* and 37.2% and 12.5% of *Salmonella* spp. isolated from retail chicken meat and LBM sewage, respectively. Molecular analysis revealed the presence of ESBL and CR genes, including $bla_{CTX-M-1}$, $bla_{CTX-M-2}$, and $bla_{NDM-1}$. The $bla_{CTX-M-1}$ gene was detected at low frequencies among ESBL-*E. coli* from retail chicken meat (1.3%) and LBM sewage (3.9%), and among ESBL-*Salmonella* spp. from retail chicken meat (3.6%), while $bla_{CTX-M-2}$ was identified in a single ESBL-*E. coli* isolate from LBM sewage. Notably, $bla_{NDM-1}$ was detected in 5.2% of CR-*E. coli* and 33.6% of CR-*Salmonella* spp. from retail chicken meat. Multidrug resistance (MDR) was observed in 98.2% and 92.2% of ESBL-*E. coli*, and 97.8% and 94.4% of ESBL-*Salmonella* spp. from retail chicken meat and LBM sewage, respectively; while, 98.8% and 100% of CR-*E. coli*, and 97.5% and 87.5% of CR-*Salmonella* spp. from both types of samples, respectively were MDR. These results highlight the urgent need for strengthened antibiotic stewardship, regular surveillance, and improve biosecurity in live bird markets in Bangladesh.

**Data availability statement:** All relevant data are within the paper and its Supporting Information files.

**Funding:** This work was supported by the BAS-USDA Program in Agriculture and Life Sciences [grant number BAS-USDA PALS LS-18] and Ministry of Science and Technology of Bangladesh [grant number SRG-221102]. The funder had no role in the design of the study; in the collection, analysis, or interpretation of data; in the writing of the manuscript; or in the decision to publish the results.

**Competing interests:** The authors have declared that no competing interests exist.

## Introduction

Antimicrobial resistance (AMR) has emerged as a critical global concern in the twenty-first century in humans, animals, and the environment [1]. The rapid rise of AMR is primarily driven by inappropriate antimicrobial use, weak regulatory frameworks, and lack of awareness, which together lead to excessive or improper antibiotic use in human medicine, veterinary practice, and agriculture, particularly in low- and middle-income countries [2]. In food animal production, the excessive and in many regions, inadequately regulated use of antibiotics including their use as growth promoters combined with poor sanitation and environmental contamination from antibiotic residues, has accelerated the development and dissemination of antimicrobial resistant bacteria worldwide [2,3]. Recently, antimicrobial use in livestock and poultry production is projected to rise by nearly 67% by 2030, particularly in low- and middle-income countries, intensifying the global AMR burden [4].

Poultry production has expanded rapidly in Bangladesh, where antimicrobial stewardship and biosecurity remain suboptimal. This rapid expansion of the chicken industry, combined with uncontrolled antibiotic use, has facilitated the emergence and spread of multidrug-resistant (MDR) bacteria through the food chain [5,6]. Among MDR strains, foodborne pathogens, especially *Escherichia coli* (*E. coli*) and *Salmonella* spp. are particularly concerning due to their zoonotic potential and ability to acquire and disseminate resistance genes across animal, human, and environmental interfaces [7]. Poultry meat, a widely consumed and affordable protein source, has increasingly been identified as a reservoir of AMR *E. coli* and *Salmonella* spp., including strains resistant to critically important antimicrobials such as extended-spectrum β-lactams and carbapenems [8,9].

ESBL-producing Enterobacterales, particularly those harboring $bla_{CTX-M}$ genes, pose a significant threat to public health by compromising frontline antibiotic therapies [10]. These resistance determinants are often plasmid-mediated, facilitating horizontal gene transfer and co-selection of MDR phenotypes [10].

In parallel, carbapenem-resistant Enterobacterales (CRE) have emerged due to environmental contamination and co-selection pressure though carbapenems are not routinely used in poultry production [11,12]. The detection of carbapenemase genes such as $bla_{KPC}$, $bla_{NDM-1}$, and $bla_{OXA-48}$ in foodborne bacteria underscores the urgency of addressing AMR within a One Health framework [13].

Live bird markets (LBMs), also known as wet markets, are central to poultry trade in Bangladesh and represent critical interfaces for AMR transmission. Birds from diverse production systems converge in LBMs, where inadequate hygiene, poor waste management, and close human–animal contact create ideal conditions for the transmission of resistant bacteria [14]. Contaminated retail chicken meat and LBM sewage serve as important pathways for human exposure to ESBL-producing and carbapenem resistant *E. coli* and *Salmonella* spp. [15–17].

Although several studies in Bangladesh have reported ESBL-producing *E. coli* and *Salmonella* spp. isolated from frozen chicken meat [12,18], poultry feces, cloacal swabs [19,20], and environmental samples [21–23], these investigations were geographically limited, sample-specific, and fragmented. To date, no comprehensive

nationwide study (population study) has systematically assessed the prevalence and distribution of ESBL-producing and carbapenem resistant *E. coli* and *Salmonella* spp. in retail chicken meat and LBM sewage across all administrative divisions of Bangladesh. Therefore, an extensive study is required to have a complete picture of ESBL-producing and carbapenem resistant *E. coli* and *Salmonella* spp. from retail chicken meat and LBM sewage covering wide areas in Bangladesh. The present study aimed to provide the first comprehensive, division-wide surveillance of ESBL-producing and carbapenem-resistant *E. coli* and *Salmonella* spp. in retail chicken meat and LBM sewage in eight divisions of Bangladesh, generating critical baseline data to inform national AMR control strategies within a One Health framework.

## Methods

### Study sites and sampling procedure

A cross-sectional survey of 64 live bird markets (LBM) was conducted in across 32 upazilas in 16 districts (the second tier of administrative regions) of Bangladesh. There are eight divisions (first tier of administrative regions) in Bangladesh. A multistage random sampling technique was used to select the LBMs. Firstly, two districts from each of the eight divisions, and then two upazilas from each selected district were randomly selected (Fig 1). Finally, two LBMs were randomly selected from each upazila. LBMs included in this study were identified using an official list provided by the respective local government authority (either city corporation or municipality) responsible for market regulation. From this list, a representative selection of markets was chosen to cover diverse geographical locations and market sizes. Sample collection from these markets was conducted following standard biosafety and ethical procedures in compliance with local regulations and official guidelines. Verbal consent was taken from the mayor of the city corporation or municipality before sampling from LBMs.

The sample size was calculated using the formula, $n = Z^2 P (1-P)/d^2$, where $n$ = sample size, $P$ = expected prevalence, $d$ = required precision of 5%, $Z$ = 1.96 at 95% confidence level [24]. The expected prevalence of *E. coli* and *Salmonella* spp. in retail chicken meat used for calculating the sample size was 85% and 70%, respectively [25,26], yielding required sample size of 196 and 323. To ensure representation across LBMs and account for within-market variability, five samples were collected from each LBM, with each market considered as a cluster. This approach resulted in a total of 320 retail chicken meat samples. Although clustering effects (design effect) were not explicitly incorporated in the initial sample size calculation, sampling was done from different vendor shops in each LBM, which likely sourced chickens from different suppliers, to reflect the intra-cluster variability. In addition, a total of 64 sewage samples were collected, with one sample from each respective LBM. Prior to sample collection, informed consent was obtained from sellers of chickens.

### Collection, transportation and processing of samples

**Meat samples.** Collected meat samples were maintained cool chain in thermal boxes and transported immediately to the laboratory, and processed on the same day. The preparation of meat samples was based on the European standard ISO-16654:2001 [27]. During processing, the meat surface was sterilized by stabbing with a hot spatula and the upper portion of meat was removed carefully. Aseptically, 25 g of each meat sample was chopped into very small fine pieces, homogenized for two minutes with 225 mL of buffered peptone water (BPW), and incubated at 37°C for 18 h. Subsequently, 1 mL of each diluted meat sample was added into two separate test tubes; one containing the nutrient broth (NB) for isolation of *E. coli*, and another one containing Rappaport-Vassiliadis Soya broth (RVS) for isolation of *Salmonella* spp., and incubated further for 18 h at 37°C for selective enrichment.

**Sewage water samples.** Sewage water sample (~15 mL) was taken aseptically in a sterile falcon tube. Bacteria were pelleted by centrifugation at 600 × g for 20 min, and re-suspended in 1 mL of BPW and incubated at 37°C for 18 h for pre-enrichment of bacteria. After that, 1 mL of this suspension was incubated in NB and RVS cultures by using broth enrichment as described for the meat samples.

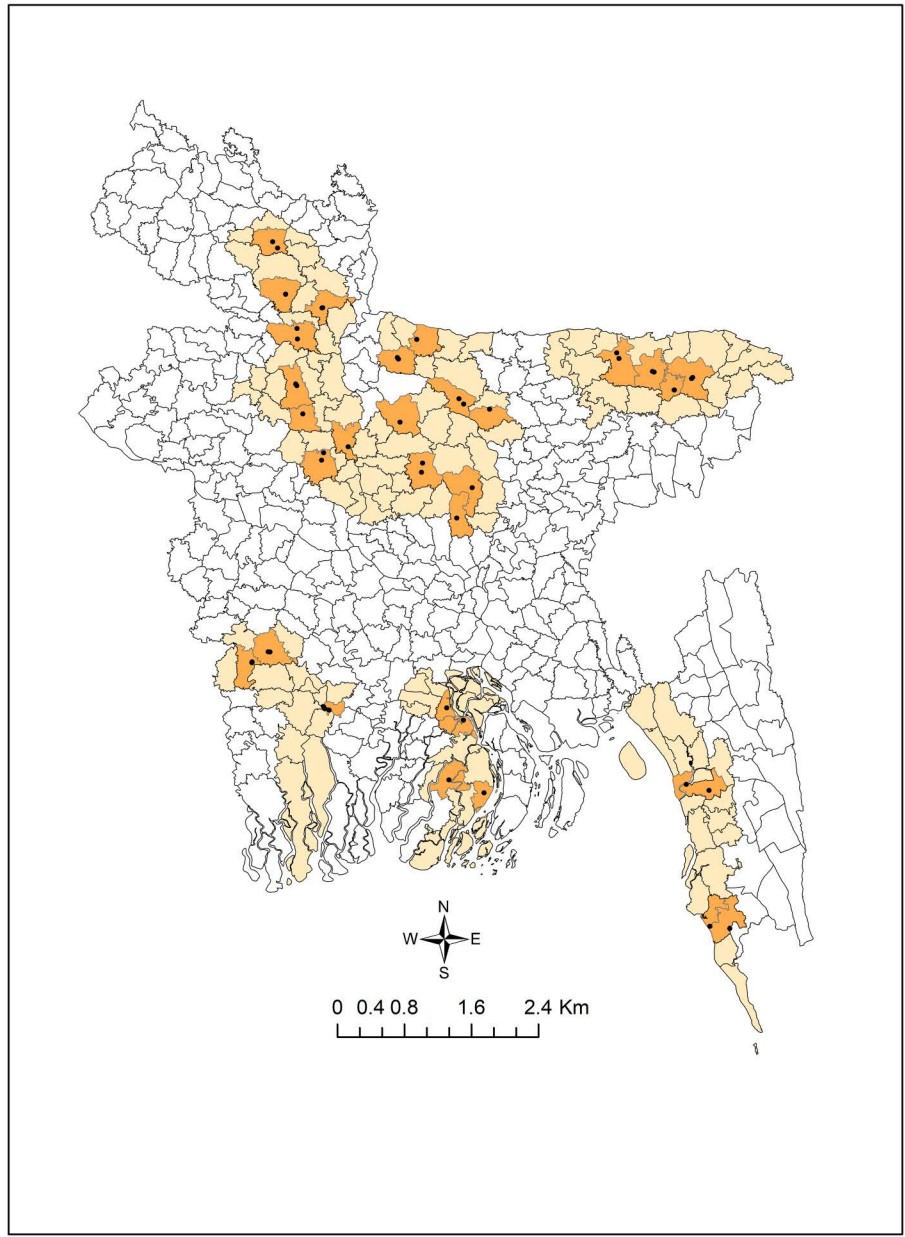

**Fig 1. Map showing selected live bird markets (•) in 32 upazilas (orange highlighted areas) of 16 districts (light yellow highlighted areas) of Bangladesh.** [The map was generated by using open-source QGIS 3.34.8 'Prizren' for desktop (qgis.org); shape files of administrative boundaries of Bangladesh were downloaded from the website of Humanitarian Data Exchange (https://data.humdata.org/dataset/cod-ab-bgd)].

## Bacteriological culture, isolation, and confirmation

After selective enrichment, a loopful of NB culture was streaked onto Eosin Methylene Blue (EMB) agar in duplicate for the isolation of *E. coli*, while a loopful of the RVS culture was streaked onto Xylose-Lysine-Deoxycholate (XLD) agar in duplicate for the isolation of *Salmonella* spp., and incubated at 37°C for 18–24 h. Typical colonies having dark blue color with a characteristic metallic sheen on EMB, and a black center and a slightly transparent zone of reddish colour on XLD

were presumptive for *E. coli*, and *Salmonella* spp., respectively. Three presumptive *E. coli* and *Salmonella* spp. colonies from each selective agar plate were picked, and then subcultured to obtain a pure culture. Gram staining and biochemical tests such as catalase, oxidase, indole, methyl red, Voges–Proskauer tests, a sugar fermentation test using triple sugar iron agar were performed from the pure culture.

Biochemically positive *E. coli* and *Salmonella* spp. isolates were then subjected to DNA extraction using the boiling method for confirmation of the bacteria by polymerase chain reaction (PCR) assay [28], and it was quantified using nanodrop spectrometer (NanoDrop One, Thermo Fisher Scientific, USA). Two uniplex PCR targeting *malB* promoter gene and *ITS* gene were used for the confirmation of *E. coli* and *Salmonella* spp. genus, respectively. Primers used for *E. coli* were ECO-1 (5'-GACCTCGGTTTAGTTCACAGA-3') and ECO-2 (5'-CACACGCTGACGCTGACCA-3') for the amplification of 585 bp [29]. The oligonucleotide primer sequences used for *Salmonella* spp. were ITSF (5'- TATAGCCCCATCGTGTAGTCAGAAC-3') and ITSR (5'-TGCGGCTGGATCACCTCCTT-3') with fragment size 312 bp [30]. Amplification reactions and PCR conditions were used in accordance to earlier study [12,18]. Positive (containing strains with reference *E. coli* and *Salmonella* spp.) and negative (sterile phosphate buffer saline) controls were included in each run. PCR amplicons were visualized by using a UV transilluminator and were photographed after electrophoresis on 1.5% agarose gel containing 0.5 µg/mL of ethidium bromide. After PCR confirmation, isolates of *E. coli* and *Salmonella* spp. were stored at –20°C with 50% (v/v) glycerol for further testing.

## Screening of extended-spectrum β-lactamases (ESBL)

Screening of ESBL production for all PCR positive *E. coli* and *Salmonella* spp. isolates were performed by the double disk synergy test [31] employing commercially available four antibiotic disks, namely amoxicillin-clavulanic acid (30 µg), cefotaxime (30 µg), ceftazidime (30 µg), and ceftriaxone (30 µg), which were placed 30 mm apart with amoxicillin-clavulanic acid disk in the middle. After overnight incubation at 37°C, the phenotypic presence of ESBL-producing *E. coli* and *Salmonella* spp. was assessed when the zone of inhibition of any of the extended-spectrum β-lactam disk expanded by at least 5 mm close to amoxicillin-clavulanic acid disk. *E. coli* ATCC 25922 and *Salmonella* Enteritidis ATCC 13076 were used as quality control bacteria.

The presence of ESBLs-encoding genes ($bla_{CTX-M-1}$, and $bla_{CTX-M-2}$) in ESBL-producing *E. coli* and *Salmonella* spp. isolates was determined by duplex PCR using specific oligonucleotide primers [32]. Primers used for $bla_{CTX-M-1}$: ctxm1-15F (5'-GAATTAGAGCGGCAGTCGGG-3') and ctxm1-02R (5'- CACAACCCAGGAAGCAGGC-3') for the amplification of 588 bp; for $bla_{CTX-M-2}$: ctxm2-39F (5'-GATGGCGACGCTACCCC-3') and ctxm2-45R (5'- CAAGCCGACCTCCCGAAC-3') for the amplification of 107 bp. The PCR reaction mixtures were prepared from 12.5 µL OneTaq® Quick-Load® 2X PCR Master Mix with standard buffer (New England, BioLabs Inc.), 1 µL of each of the forward and reverse primers with a concentration of 10 pmol, 1 µL template DNA from each isolate, and 7.5 µL of nuclease-free water to reach a total volume of 25 µL. The PCR conditions used were as follows: initial denaturation at 95°C for 5 min, followed by 25 cycles of denaturation at 95°C for 30 s, annealing at 60°C for 1 min, and extension at 72°C for 1 min, with a final extension at 72°C for 10 min. Positive controls consisted of previously characterized reference isolates harboring $bla_{CTX-M-1}$ and $bla_{CTX-M-2}$ genes [12,18], while sterile phosphate-buffered saline served as the negative control in each PCR run; then the PCR products were electrophoresed on a 1.5% agarose gel.

## Screening of carbapenem-resistant isolates

Carbapenem resistance in *E. coli* and *Salmonella* spp. was screened by the disk diffusion method on Mueller-Hinton agar plates using commercially available imipenem (10 µg) and meropenem (10 µg) disk (Biomaxima, Lublin, Lubelskie, Poland). Isolates that were resistant to at least one carbapenem drug were considered as CR isolates [31,33], and selected for the detection of the carbapenemase genes using PCR.

The presence of the carbapenemase-encoding $bla_{NDM-1}$ gene was assessed by uniplex PCR employing gene-specific oligonucleotide primers as described earlier [34]. The primer sequences were NDM-1 forward: 5'-CTTCCAACGGTTTGATCGTC-3'

and NDM-1 reverse: 5′-TAGTGCTCAGTGTCGGCATC-3′ with a fragment size of 465 bp. The PCR reaction mixtures, with a total volume of 25 µL, contained 12.5 µL of OneTaq® Quick-Load® 2X PCR Master Mix with standard buffer (New England, BioLabs Inc.), 8.5 µL of water, 1 µL of template DNA from each isolate, and 1.5 µL of each primer at a concentration of 15 pmol. PCR was run with an initial denaturation at 95°C for 7 min; 30 cycles of 95°C for 1 min, 55°C for 1 min, and 68°C for 1 min; and a final elongation at 68°C for 7 min. The PCR product was electrophoresed on a 1.5% agarose gel to determine the size of the product. Positive controls consisted of previously characterized reference isolates harboring $bla_{NDM-1}$ gene [12], while sterile phosphate-buffered saline served as the negative control in each PCR run. In addition, a 100 bp DNA ladder (Size range: 100–1000 bp, New England, BioLabs Inc.) was run simultaneously to detect the size of the bands.

## Antimicrobial susceptibility testing

The antimicrobial susceptibility testing of all ESBL- producing and CR-*E. coli* and *Salmonella* spp. isolates was carried out against 29 different antimicrobials representing 14 distinct antimicrobial classes by Kirby-Bauer disk diffusion method according to the guidelines recommended by the Clinical and Laboratory Standard Institute (CLSI) [31]. For colistin and polymyxin B, minimum inhibitory concentrations (MICs) were determined by broth microdilution method. The interpretation of the results was based on the guidelines of CLSI [31], and in some cases, when breakpoints of some antimicrobials (pefloxacin, cephalexin, cephradine, colistin, polymyxin B) in CLSI were unavailable, the guideline of European Committee on Antimicrobial Susceptibility Testing (EUCAST) was used [35]. Isolates resistant to at least one agent in three or more antimicrobial classes were defined as multidrug resistant (MDR), while isolates resistant to at least one agent in all but two or fewer antimicrobial classes, i.e., bacterial isolates remain susceptible to only one or two classes were defined as extensively drug resistant (XDR) [36]. Moreover, when isolates remained susceptible to three classes were defined as possible extensively drug resistant (pXDR) as per outline given in the report of Magiorakos, Srinivasan (36). The following antimicrobials were tested: penicillins [ampicillin (10 µg)], penicillins + β-lactamase inhibitors [amoxicillin-clavulanic acid (30 µg)], antipseudomonal penicillins + β-lactamase inhibitors [pipercillin-tazobactam (110 µg)], non-extended spectrum cephalosporins [first-generation cephalosporins: cephalexin (30 µg), cephradine (30 µg); second-generation cephalosporins: cefuroxime (30 µg), cefaclor (30 µg)], extended-spectrum cephalosporins [third-generation cephalosporins: cefotaxime (30 µg), ceftriaxone (30 µg), ceftazidime (30 µg), cefixime (5 µg); fourth-generation cephalosporins: cefepime (30 µg)], monobactams [aztreonam (30 µg)], cephamycins [cefoxitin (30 µg)], fluoroquinolones [nalidixic acid (30 µg), ciprofloxacin (5 µg), levofloxacin (5 µg), norfloxacin (10 µg), ofloxacin (5 µg), gatifloxacin (5 µg), pefloxacin (5 µg)], tetracyclines [doxycycline (10 µg)], aminoglycosides [gentamicin (10 µg), amikacin (30 µg)], folate pathway inhibitors [trimethoprim-sulfamethoxazole (25 µg)], glycylcyclines [tigecycline (15 µg)], phenicols [chloramphenicol (30 µg)], polymyxins [colistin, polymyxin B].

## Data analysis

Descriptive statistics were used to compute the prevalence of bacteria and phenotypic resistance percentage. The significant differences in prevalence of bacteria and resistance percentage among sample types, and study areas were determined by using chi-square test (Z-test for proportions) and Fisher's exact test (if one or more expected cell frequencies were less than 5). The statistical software package SPSS version 22.0 (IBM, Somers, NY) was used for the analyses.

## Spatial mapping

The spatial mapping was done with district-wise prevalence data of ESBL-*E. coli* and *Salmonella* spp. from 64 live bird markets using open-source QGIS 3.34.8 'Prizren' for desktop (qgis.org). For mapping, the shape files of administrative boundaries of Bangladesh were downloaded from the website of Humanitarian Data Exchange (https://data.humdata.org/dataset/cod-ab-bgd).

 

## Ethics statement

This study was approved by the Animal Welfare and Experimentation Ethics Committee of Bangladesh Agricultural University, Mymensingh. The approval number was AWEEC/BAU/2017(13). No animal experimentation was done in this study. However, informed written consent was taken from the live bird shop owners before sampling.

## Results

### Prevalence of ESBL-producing and carbapenem-resistant bacteria

The prevalence of ESBL-producing *E. coli* was 70% (224/320, 95% CI: 65–75%) in retail chicken meat and 79.7% (51/64, 95% CI: 68–89%) in LBM sewage samples. In comparison, ESBL-producing *Salmonella* spp. were detected in 43.4% (139/320, 95% CI: 38–49%) of retail chicken meat samples and 28.1% (18/64, 95% CI: 18–41%) of LBM sewage samples. Carbapenem-resistant (CR) *E. coli* in were identified in 54.1% (173/320, 95% CI: 48–60%) of retail chicken meat and 46.9% (30/64, 95% CI: 34–60%) of LBM sewage samples (Table 1). By contrast, the prevalence of CR-*Salmonella* spp. was lower, at 37.2% (119/320, 95% CI: 32–43%) in retail chicken meat and 12.5% (8/64, 95% CI: 6–23%) in LBM sewage samples.

### Spatial distribution of ESBL-producing and CR-*E. coli* and *Salmonella* spp.

District-wise analysis of retail chicken meat samples showed that the prevalence of ESBL-producing *E. coli* was highest in Barishal and Sylhet districts (90%, 95% CI: 68–99%), followed by Patuakhali and Jashore (85%, 95% CI: 62–97%), and Mymensingh, Bogura and Chattogram (80%, 95% CI: 56–94%) compared with other districts (Fig 2a, S1 Table). In contrast, among ESBL-producing *Salmonella* spp., the highest prevalence was observed in Sylhet and Gazipur districts (70%, 95% CI: 46–88%), while the lowest prevalence was recorded in Khulna and Sunamganj districts (20%, 95% CI: 6–44%) and 25%, 95% CI: 9–49%, respectively) (Fig 2b, S1 Table). CR-*E. coli* was most prevalent in Barishal district (80%, 95% CI: 56–94%), followed by Mymensingh and Bogura (75%, 95% CI: 51–91%) and Jashore (70%, 95% CI: 46–88%) districts, with lower prevalence in Cox's Bazar and Rangpur districts (30%, 95% CI: 12–54%) (Fig 2c, S1 Table). Conversely, CR-*Salmonella* spp. showed the highest occurrence in Rangpur and Gaibandha districts (65%, 95% CI: 41–85%), followed by Cox's Bazar (60%, 95% CI: 36–81%), with the lowest prevalence in Sylhet district (5%, 95% CI: 0.1–25%) (Fig 2d, S1 Table).

Analysis of LBM sewage samples showed a high prevalence (100%, 95% CI: 40–100%) of ESBL-*E. coli* observed in Mymensingh, Sherpur, Bogura, Sirajganj, Gazipur, Patuakhali, Sylhet, Khulna, and Jashore districts. In contrast, a lower prevalence was detected in Chattogram (25%, 95% CI: 1–81%), and no ESBL-producing *E. coli* were recovered from Rangpur district (S1 Fig, S1 Table). Similarly, ESBL-producing *Salmonella* spp. were most prevalent in Bogura district (75%, 95% CI: 19–99%), whereas lower prevalence (25%, 95% CI: 1–81%) was observed in Rangpur, Sirajganj, Tangail, Sunamganj, and Sherpur districts; none were detected in Chattogram, Cox's Bazar, Patuakhali, Khulna, and Jashore districts (S1 Fig, S1 Table).

**Table 1. Prevalence of ESBL-producing and carbapenem-resistant *E. coli* and *Salmonella* spp. isolated from retail chicken meat and LBM sewage samples.**

| ESBL-producing and CR bacteria | Retail chicken meat (n = 320) | | LBM sewage (n = 64) | |
|---|---|---|---|---|
| | No. (%) of isolates | 95% CI | No. (%) of isolates | 95% CI |
| ESBL-*E. coli* | 224 (70)[a] | 65-75 | 51 (79.7)[a] | 68-89 |
| ESBL-*Salmonella* spp. | 139 (43.4)[a] | 38-49 | 18 (28.1)[a] | 18-41 |
| CR-*E. coli* | 173 (54.1)[a] | 48-60 | 30 (46.9)[a] | 34-60 |
| CR-*Salmonella* spp. | 119 (37.2)[a] | 32-43 | 8 (12.5)[b] | 6-23 |

[a,b]Values in the same row with different superscripts differ significantly between sample types (p ≤ 0.05).

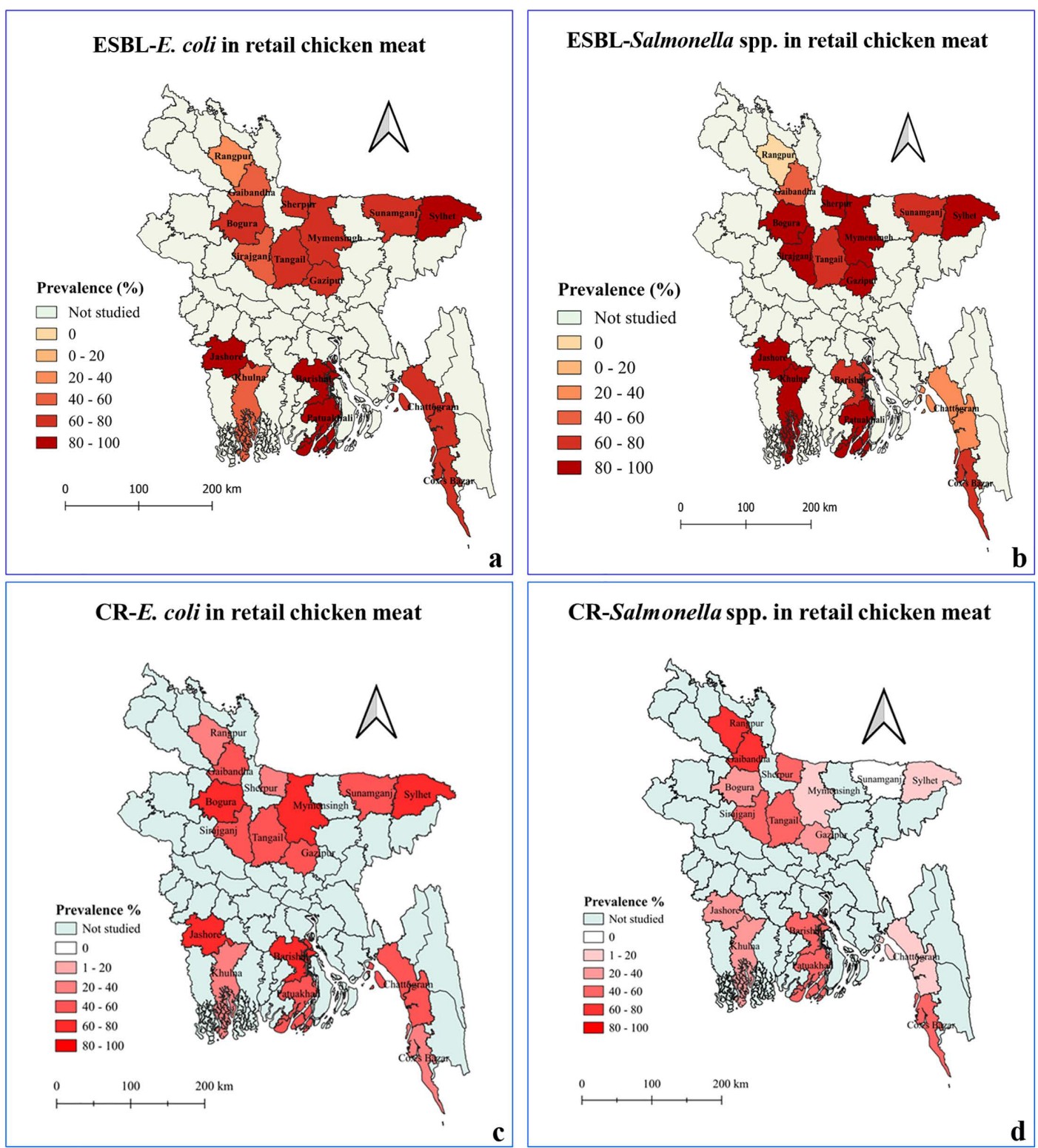

**Fig 2. Maps showing district-wise distribution of ESBL-*E. coli*, ESBL-*Salmonella* spp., CR-*E. coli*, and CR-*Salmonella* spp. isolated from retail chicken meat.**

The prevalence of CR-*E. coli* was highest in Mymensingh district (100%, 95% CI: 40–100%) and lowest in Sirajganj district (25%, 95% CI: 1–81%). CR-*Salmonella* spp. were most frequently detected in Barishal and Gaibandha districts (50%, 95% CI: 7–93%) compared with other districts (S1 Fig, S1 Table).

## Prevalence of ESBL-encoding and carbapenemase genes

Among ESBL-encoding genes, three (1.3%, 95% CI: 0.3−4%) isolates of ESBL-*E. coli* from retail chicken meat and 2 (3.9%, 95% CI: 0.5−13%) isolates from LBM sewage samples harbored the $bla_{CTX-M-1}$ gene, while 5 (3.6%, 95% CI: 1−8%) isolates of ESBL-*Salmonella* spp. from retail chicken meat were positive for $bla_{CTX-M-1}$ gene (Fig 3 and S2a Fig). The $bla_{CTX-M-2}$ gene was detected in one (2%, 95% CI: 0.1−10%) ESBL-*E. coli* isolate from LBM sewage samples. The carbapenemase gene, $bla_{NDM-1}$, was detected in 5.2% (9/173, 95% CI: 2−10%) of CR-*E. coli* isolates and 33.6% (40/119, 95% CI: 30−40%) isolates of CR-*Salmonella* spp. from retail chicken meat samples, while none of the CR-*E. coli* and CR-*Salmonella* spp. isolates from LBM sewage samples carried $bla_{NDM-1}$ gene (Fig 3 and S2b Fig).

## Multidrug resistance pattern

A high prevalence of multidrug resistance (MDR) was observed among ESBL-producing and CR *E. coli* and *Salmonella* spp. isolated from both retail chicken meat and LBM sewage. Among ESBL-producing isolates from retail chicken meat, MDR was detected in 98.2% (220/224) of *E. coli* and 97.8% (136/139) of *Salmonella* spp., while corresponding proportions in LBM sewage were 92.2% (47/51) and 94.4% (17/18), respectively (Table 2). For CR isolates, MDR prevalence was detected in 98.8% (171/173) of *E. coli* and 97.5% (116/119) of *Salmonella* spp. from retail chicken meat, while all *E. coli* isolates (100%, 30/30) and 87.5% (7/8) of *Salmonella* spp. from LBM sewage samples were MDR (Table 3).

Notably, it was observed that two isolates (0.9%) of ESBL-*E. coli*, 4 isolates (2.9%) of ESBL-*Salmonella* spp., 2 (1.2%) isolates of CR-*E. coli*, and 2 (1.7%) isolates of CR-*Salmonella* spp. from retail chicken meat were possible extensively drug resistant (pXDR) (Table 2-3).

## Individual antimicrobial resistance pattern

The highest percentage of ESBL-*E. coli* (40.6%) and *Salmonella* spp. (34.5%) as well as CR-*E. coli* (42.8%) and *Salmonella* spp. (42.9%) isolates recovered from retail chicken meat exhibited resistant to 9–12 antimicrobial agents (Fig 4). Approximately one-fourth (~20%) of ESBL- and CR-producing *E. coli* and *Salmonella* spp. were resistant to 5–8

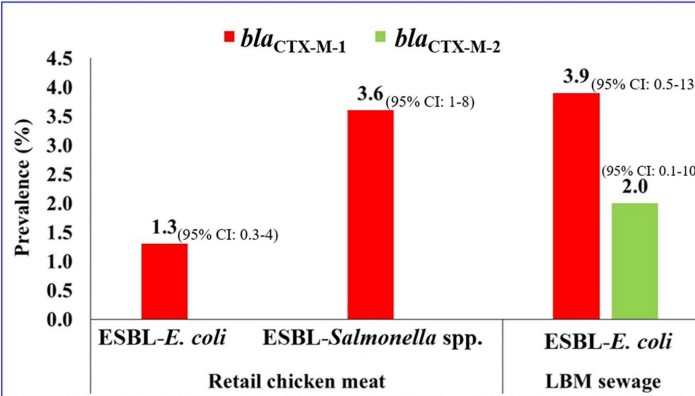
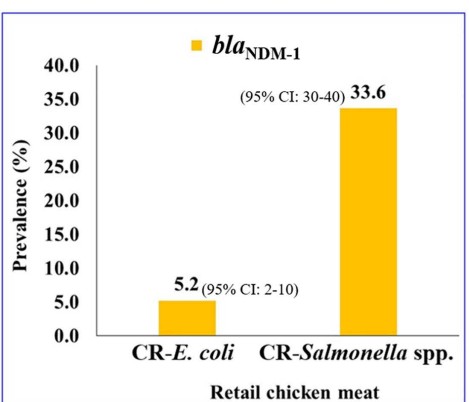

**Fig 3. Prevalence of resistance genes in ESBL-producing and carbapenem-resistant *E. coli* and *Salmonella* spp. isolated from retail chicken meat and LBM sewage samples.**

**Table 2. Multidrug resistance patterns observed among ESBL-*E. coli* and *Salmonella* spp. isolated from retail chicken meat and LBM sewage samples.**

| Antimicrobial classes | No. (%) of isolates | | | |
|---|---|---|---|---|
| | Retail chicken meat | | LBM sewage | |
| | ESBL-*E. coli* (n=224) | ESBL-*Salmonella* spp. (n=139) | ESBL-*E. coli* (n=51) | ESBL-*Salmonella* spp. (n=18) |
| 3-5 | 88 (39.3)[a] | 35 (25.2)[a] | 21 (41.2)[a] | 6 (33.3)[a] |
| 6-8 | 117 (52.2)[a] | 66 (47.5)[b] | 22 (43.1)[a] | 11 (61.1)[b] |
| ≥ 9 | 15 (6.7)[b] | 35 (25.2)[a] | 4 (7.8)[b] | 0 |
| **MDR** | **220 (98.2)** | **136 (97.8)** | **47 (92.2)** | **17 (94.4)** |
| **pXDR** | **2 (0.9)** | **4 (2.9)** | **0** | **0** |

[ab]Values in the same column with different superscripts differ significantly (p ≤ 0.05).

**Table 3. Multidrug resistance patterns observed among CR-*E. coli* and *Salmonella* spp. isolated from retail chicken meat and LBM sewage samples.**

| Antimicrobial classes | No. (%) of isolates | | | |
|---|---|---|---|---|
| | Retail chicken meat | | LBM sewage | |
| | CR-*E. coli* (n=173) | CR-*Salmonella* spp. (n=119) | CR-*E. coli* (n=30) | CR-*Salmonella* spp. (n=8) |
| 3-5 | 65 (37.6)[a] | 37 (31.1)[a] | 12 (40.0)[a] | 2 (25.0)[a] |
| 6-8 | 92 (53.2)[a] | 63 (52.9)[b] | 15 (50.0)[a] | 5 (62.5)[b] |
| ≥ 9 | 14 (8.1)[b] | 16 (13.4)[c] | 3 (10.0)[b] | 0 |
| **MDR** | **171 (98.8)** | **116 (97.5)** | **30 (100.0)** | **7 (87.5)** |
| **pXDR** | **2 (1.2)** | **2 (1.7)** | **0** | **0** |

[abc]Values in the same column with different superscripts differ significantly (p ≤ 0.05).

antimicrobial agents. Resistance to ≥13 antimicrobial agents was also common, affecting 29.0% (65/224) of ESBL-*E. coli*, 38.1% (53/139) of ESBL-*Salmonella* spp., 30.6% (53/173) of CR-*E. coli*, and 28.6% (34/119) of CR-*Salmonella* spp.

Among isolates from LBM sewage samples, resistance to 9–12 antimicrobial agents was observed in 27.5% of ESBL-*E. coli* and 30% of CR-*E. coli*, while resistance to 5–8 antimicrobials recorded in 23.5% and 16.7%, respectively (Fig 4). In contrast, most ESBL-*Salmonella* spp. isolates (38.9%) exhibited resistance to 5–8 antimicrobial agents, whereas CR-*Salmonella* spp. isolates (37.5%) showed resistance to 9–12 agents. Resistance to ≥13 antimicrobial agents was observed in 27.5% of ESBL-*E. coli*, 5.6% of *Salmonella* spp., 43.3% of CR-*E. coli*, and 12.5% of CR-*Salmonella* spp. isolates.

The heat map analysis showed that ESBL-*E. coli* from retail chicken meat exhibited the highest resistance to pefloxacin (98.7%), followed by ampicillin (95.5%), trimethoprim-sulfamethoxazole (92.0%), nalidixic acid (79.9%), and doxycycline (75.4%) (Fig 5, S2 Table). Moderate resistance was observed to fluoroquinolones, including ciprofloxacin (67.4%), ofloxacin (67.0%), gatifloxacin (66.1%), and levofloxacin (61.6%), whereas resistance to tigecycline (0.4%) and cefoxitin (2.7%) was lowest. Resistance to colistin was detected in 39% of isolates. In ESBL-*Salmonella* spp. from retail chicken meat, the highest resistance was observed to colistin, polymyxin B, and doxycycline (98.6% each), followed by pefloxacin (89.9%), trimethoprim-sulfamethoxazole (89.2%), ampicillin (84.9%), and nalidixic acid (80.6%). In contrast, resistance to tigecycline (1.4%), piperacillin-tazobactam (4.3%), cefoxitin (9.4%), and aztreonam (11.5%) remained low.

ESBL-*E. coli* from LBM sewage showed similarly high resistance to pefloxacin (94.1%), trimethoprim-sulfamethoxazole (86.3%), ampicillin (82.4%), amoxicillin-clavulanic acid (62.7%), and doxycycline (60.8%) with low resistance to cefoxitin (2.0%), piperacillin-tazobactam (3.9%), ceftazidime and polymyxin B (7.8% each), and amikacin (9.8%) (Fig 5, S2 Table). Colistin resistance detected in 37.3% of isolates, while all isolates were susceptible to tigecycline. ESBL-*Salmonella* spp.

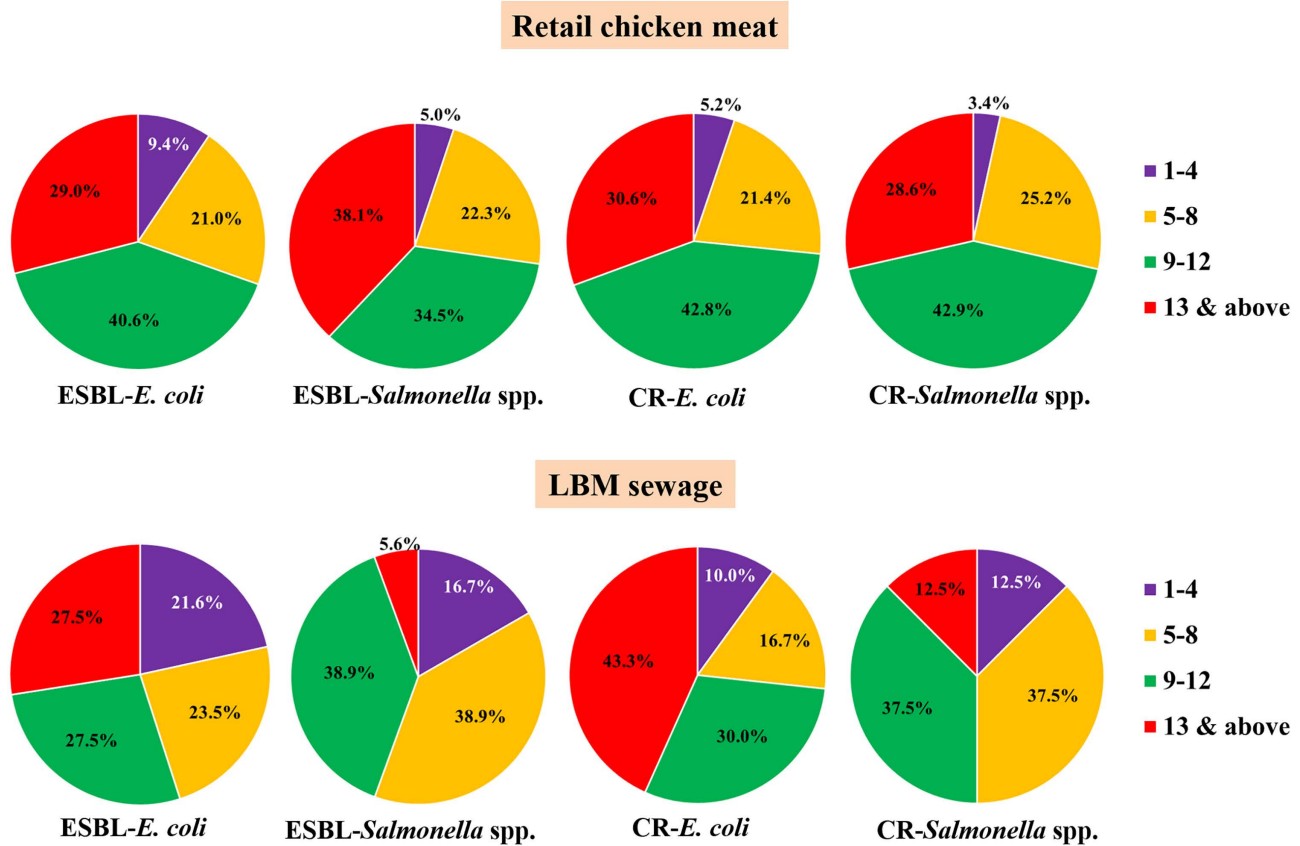

**Fig 4. Resistance distribution of ESBL-producing and CR -*E. coli* and *Salmonella* spp. to number of antimicrobial agents isolated from retail chicken meat and LBM sewage samples.**

from LBM sewage were predominantly resistant to colistin and polymyxin B (94.4%), doxycycline and pefloxacin (83.3% each), and trimethoprim-sulfamethoxazole (72.2%), whereas resistance to levofloxacin, ceftazidime, and cefoxitin was minimal (5.6%), and no resistance was detected to several cephalosporins, piperacillin-tazobactam, amikacin, or tigecycline.

CR-*E. coli* from both retail chicken meat and LBM sewage exhibited high resistance to pefloxacin (100% each), ampicillin (96.0% and 86.7%), trimethoprim-sulfamethoxazole (93.1% and 90.0%), nalidixic acid (84.4% and 76.7%), and doxycycline (75.7% and 73.3%), respectively (Fig 5, S2 Table). For CR-*Salmonella* spp., nearly all isolates were resistant to colistin and polymyxin B (99.2% and 100%), pefloxacin (99.2% and 87.5%), doxycycline (95.8% and 87.5%), nalidixic acid (89.1% and 75.0%), trimethoprim-sulfamethoxazole (88.2% and 62.5%), and ampicillin (80.7% and 62.5%) from retail chicken meat and LBM sewage, respectively.

## Discussion

The present LBM survey documented a comprehensive findings on the magnitude and distribution of ESBL-producing and CR-*E. coli* and *Salmonella* spp. alongside their antimicrobial resistance patterns in retail chicken meat and LBM sewage samples across 16 districts from all eight divisions of Bangladesh. The high prevalence of ESBL-producing *E. coli* (70%) and *Salmonella* spp. (43.4%) in retail chicken meat exceeds reports from India, Turkiye, Taiwan, and South Korea [15,37–39], indicating a subtantial food safety risk to consumers. Similarly, the prevalence of ESBL-producing *E. coli* and *Salmonella* spp. was very high (79.7% and 28.1%, respectively) in LBM sewage samples. The markedly higher

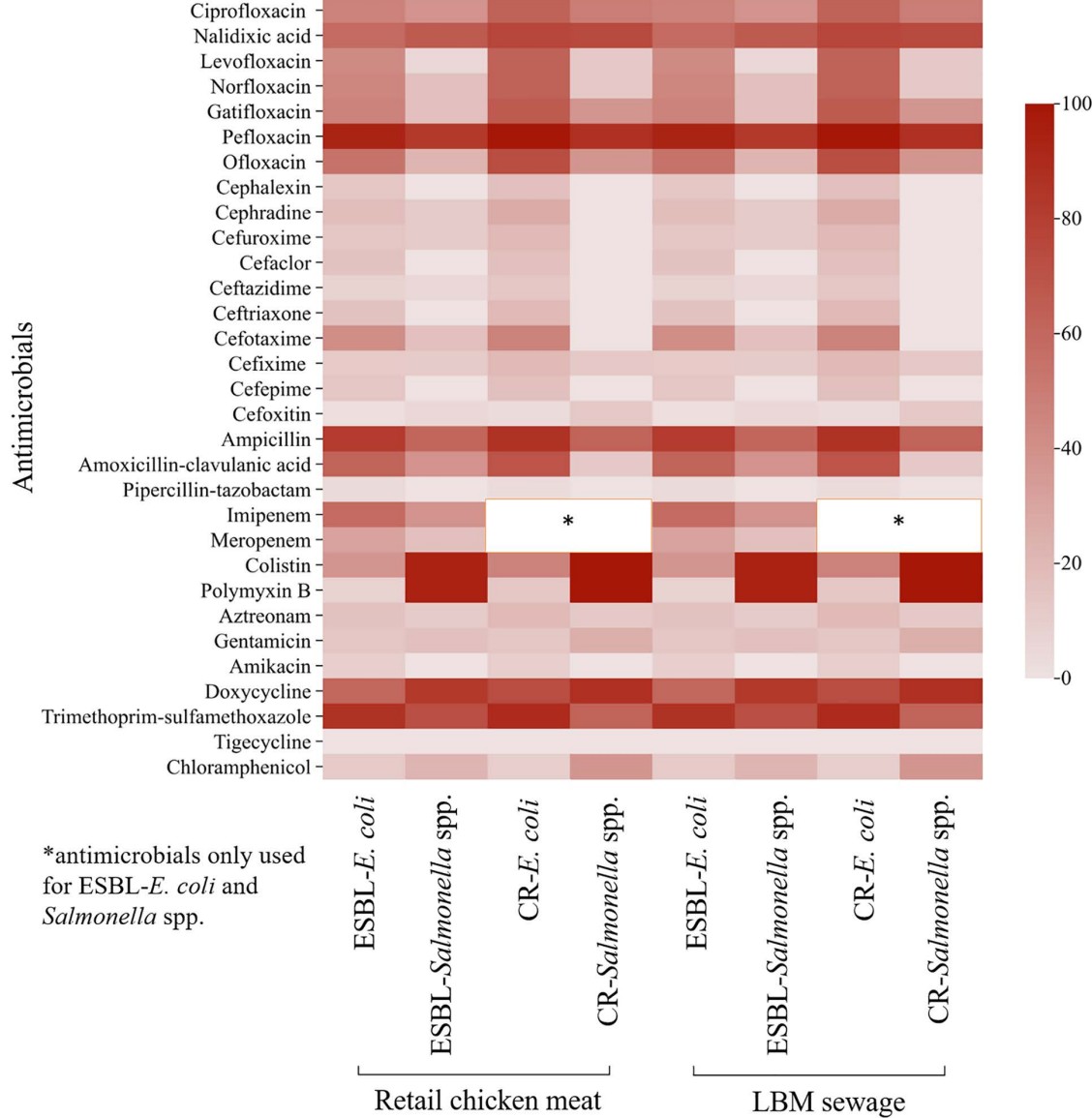

**Fig 5. Heat map showing individual antimicrobial resistance pattern of ESBL-*E. coli*, ESBL-*Salmonella* spp., CR-*E. coli* and CR-*Salmonella* spp. isolated from retail chicken meat and LBM sewage samples.**

prevalence in LBM sewage samples suggest that LBM function as environmental reservoirs and amplification points for resistant bacteria, likely driven by waste accumulation and poor hygiene practices [40].

Concerning carbapenem resistance, our investigation identified 54.1% CR-*E. coli* in retail chicken meat, and 46.9% in LBM sewage samples. On the other hand, CR-*Salmonella* spp. were detected in 37.2% and 12.5% of retail chicken meat and LBM sewage samples, respectively. These findings are alarming because carbapenems are regarded as essential antibiotics reserved for the treatment of severe infections caused by MDR organisms in human medicine, and the presence of CR foodborne pathogens in the poultry value chain represents a significant risk to public health [13]. The identification of the carbapenemase gene $bla_{NDM-1}$ in CR-*E. coli* and, notably, in a high proportion of CR-*Salmonella* spp.

from retail chicken meat represents an alarming finding. The $bla_{NDM-1}$ gene mediates resistance against a broad range of β-lactam antibiotics, notably carbapenems that are considered critical last-line agents in human healthcare [13]. The presence of this gene in retail meat provides strong evidence of foodborne dissemination of critically important resistance determinants. Its absence in LBM sewage may suggest differing selective pressures or contamination pathways in meat processing environments compared with wastewater systems; however, this interpretation remains speculative.

The detection of ESBL-encoding genes, particularly $bla_{CTX-M-1}$, in both retail chicken meat and LBM sewage provides direct evidence of the circulation of globally important resistance determinants within the poultry value chain. Although the prevalence of $bla_{CTX-M-1}$ in this study was lower than that reported in Turkey and South Korea [41,42], its presence in food products remains concerning due to the potential for transmission to humans through food handling and consumption [43]. The limited detection of $bla_{CTX-M-2}$ suggests a more restricted distribution of this gene variant in the studied environment.

The higher prevalence of ESBL-producing and CR-*E. coli* and *Salmonella* spp. in retail chicken meat and LBM sewage could be influenced by contamination during slaughtering, processing, transportation, and retail handling, which may promote cross-contamination and bacterial multiplication [44]. Similarly, the observed district-level variation in prevalence may reflect regional differences in poultry farming practices, antimicrobial use, and hygiene standards, consistent with earlier reports from different regions of the country [5,12,18–21].

Of particular concern is the extremely high proportion of MDR isolates detected in both retail chicken meat and LBM sewage samples. More than 90% of ESBL-*E. coli* and *Salmonella* spp. from retail chicken meat and LBM sewage samples, and nearly all CR-*E. coli* and *Salmonella* spp. from retail chicken meat (~98%), and 100% of CR-*E. coli* and 87.5% of CR-*Salmonella* spp. from LBM sewage samples were MDR.These findings are consistent with reports from other low- and middle-income countries [17,45]. Studies from major wholesale and urban LBMs in Bangladesh have documented similarly elevated MDR rates, indicating that such markets are hotspots for resistant bacteria across the poultry supply chain [20,46].The detection of MDR strains in LBM sewage likely reflects characteristic market practices, including on-site slaughtering and processing of poultry, inadequate hygiene and sanitation, mixing of poultry waste with wash water, and the direct discharge of untreated effluents into municipal drains. Such conditions facilitate the accumulation, persistence, and exchange of resistant bacteria and resistance genes. The detection of MDR strains in sewage provides clear evidence that resistant bacteria and antimicrobial resistance genes are widely circulating within the community and entering environmental reservoirs [47,48]. Sewage systems are well recognized as major conduits for AMR dissemination through horizontal gene transfer, particularly in settings with inadequate wastewater treatment infrastructure [49]. Systematic reviews of poultry environments across Bangladesh similarly estimate that MDR *E. coli* occurs in over 90% of isolates from poultry and poultry market settings, with resistance spanning many antimicrobial classes, including critically important drugs [50]. Of note, the current study also observed that 0.9% of ESBL-*E. coli*, 2.9% of ESBL-*Salmonella* spp., 1.2% of CR-*E. coli* and 1.7% of CR-*Salmonella* spp. in retail chicken meat were pXDR. A study from Pakistan reported that the presence of an extensively drug resistant *Salmonella enterica* in retail chicken meat [51]. The high levels of MDR and the existence of pXDR among ESBL-producing and CR-*E. coli* and *Salmonella* spp. may be linked to the widespread and largely unrestricted use of antibiotics in poultry production in Bangladesh. In addition, inadequate waste management systems likelyincrease selective pressure for MDR bacteria, facilitating the emergence and dissemination of MDR foodborne bacteria throughout the poultry production chain and into retail poultry products [52]. To reduce MDR foodborne bacterial contamination in retail chicken meat, maintaining adequate hygiene during and after chicken slaughter is very essential.

The resistance profiles observed, particularly high resistance to fluoroquinolones, ampicillin, trimethoprim–sulfamethoxazole, doxycycline, and colistin, are consistent with the known widespread use of these inexpensive and readily available antimicrobials in poultry production [3,15,37,38,53,54]. The high resistance to colistin likely reflects historical use overtime in Bangladesh despite regulatory bans on import and manufacture of colistin for use in poultry production since 2019 [55]. In contrast, the relatively low resistance to tigecycline, piperacillin-tazobactam and cefoxitin is consistent with their limited use in veterinary practice in Bangladesh.

This high resistance rates of ESBL-*E. coli* and *Salmonella* spp. in retail chicken meat as well as in LBM sewage samples reflect its widespread use in poultry feed in Bangladesh. Recently, Bangladesh government banned the manufacture, sale and distribution of colistin and its formulations for poultry production to tackle colistin resistance The findings highlight the urgent need for targeted interventions, particularly improved antibiotic management and enhanced sanitation services in LBMs. Bangladesh already has regulatory structures through the Directorate General of Drug Administration and the Department of Livestock Services, providing a feasible foundation for enforcing prescription-based antimicrobial use and restricting critically important antibiotics. While challenges remain due to informal poultry production systems and limited veterinary coverage, incremental measures such as farmer education, vaccination programs, strengthened farm biosecurity, and increased veterinary oversight are practical and cost-effective approaches to reducing antimicrobial dependence.

Improving LBM sanitation is similarly feasible through pragmatic, low-cost interventions, including routine cleaning and disinfection, improved waste disposal and drainage, segregation of clean and contaminated zones, and periodic market rest days. These measures can be implemented through collaboration among local authorities, market committees, and vendors without requiring major infrastructural investment. Evidence from comparable settings suggests that such interventions can significantly reduce environmental contamination and bacterial load.

It would be worthwhile if samples were taken from slaughterer or workers. However, retail chicken meat samples were collected from 64 LBMs of 32 upazilas of 16 districts in all the divisions of Bangladesh; thus, the data represent the scenario of whole Bangladesh. Further work should be performed to characterize ESBL-producing and CR-*E. coli* and *Salmonella* spp. isolates of poultry and human origin from the same sites sharing the same resistance markers in order to highlight potential horizontal gene transfer by these foodborne organisms.

## Conclusion

The results highlight the role of contaminated retail chicken meat as a potential source of ESBL-producing and CR-*E. coli* and *Salmonella* spp, which may disseminate to humans and the environment. These results raise serious concerns regarding public health and food safety concerns, as retail chicken meat can serve as a reservoir of MDR bacteria capable of transmission through the food chain. These findings highlight the necessity of strengthening hygienic practices and sanitary handling practices at retail markets to limit microbial contamination, alongside enhanced consumer education to promote safe handling and proper preparation of poultry products. In addition, the conclusions emphasize the strengthening of biosecurity measures throughout all stages of poultry production, from farm to market, to limit the introduction and spread of resistant pathogens.

## Supporting information

**S1 Fig. Maps showing district-wise distribution of ESBL-*E. coli*, ESBL-*Salmonella* spp., CR-*E. coli* and CR-*Salmonella* spp. isolated from LBM sewage samples.**
(TIF)

**S2 Fig. ESBL-encoding and carbapenemase genes in ESBL-producing and carbapenem-resistant *E. coli* and *Salmonella* spp. isolates by uniplex and duplex PCR following 1.5% agarose gel electrophoresis and ethidium bromide staining.** Legends: a) L = DNA marker (100 bp), Lane 1 = Positive control ($bla_{CTX-M-1}$ and $bla_{CTX-M-2}$), Lane 2 = Negative control, Lane 3 = Positive for $bla_{CTX-M-2}$ gene; Lane 4−11 = Positive for $bla_{CTX-M-1}$ gene. b) L = DNA marker (100 bp), Lane 1 = Positive control ($bla_{NDM-1}$), Lane 2 = Negative control, Lane 3−11 = Positive for $bla_{NDM-1}$.
(TIF)

**S1 Table. District-wise distribution of ESBL-*E. coli*, ESBL-*Salmonella* spp., CR-*E. coli*, and CR-*Salmonella* spp. isolated from retail chicken meat and LBM sewage samples.**
(DOCX)

**S2 Table. Individual antimicrobial resistance pattern of ESBL-*E. coli*, ESBL-*Salmonella* spp., CR-*E. coli* and CR-*Salmonella* spp. isolated from retail chicken meat and LBM sewage samples.**
(DOCX)

**S1 Raw images. Raw agarose gel electrophoresis images of PCR assays showing ESBL-encoding and carbapenemase genes in ESBL-producing and carbapenem-resistant *E. coli* and *Salmonella* spp.**
(PDF)

## Acknowledgments

The authors would like to thank the Bangabandhu Science and Technology Fellowship Trust for giving the fellowship. The authors would also like to thank the Department of Medicine, Faculty of Veterinary Science, Bangladesh Agricultural University, Mymensingh for the support during the research.

## Author contributions

**Conceptualization:** MD. TAOHIDUL ISLAM.

**Data curation:** Mst. Sonia Parvin.

**Formal analysis:** Mst. Sonia Parvin, MD. TAOHIDUL ISLAM.

**Funding acquisition:** MD. TAOHIDUL ISLAM.

**Investigation:** Mst. Sonia Parvin, MD. TAOHIDUL ISLAM.

**Methodology:** Mst. Sonia Parvin, Sudipta Talukder, Sayra Tasnin Sharmy, Md. Mehedi Hasan.

**Project administration:** MD. TAOHIDUL ISLAM.

**Resources:** Sudipta Talukder, Sayra Tasnin Sharmy, Md. Mehedi Hasan.

**Supervision:** MD. TAOHIDUL ISLAM.

**Validation:** MD. TAOHIDUL ISLAM.

**Writing – original draft:** Mst. Sonia Parvin.

**Writing – review & editing:** Sudipta Talukder, Sayra Tasnin Sharmy, Md. Mehedi Hasan, MD. TAOHIDUL ISLAM.

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
