## [Decision Letter · Decision Letter 0]

23 Nov 2025

Please submit your revised manuscript by Jan 07 2026 11:59PM. If you will need more time than this to complete your revisions, please reply to this message or contact the journal office at plosone@plos.org. . . . A rebuttal letter that responds to each point raised by the academic editor and reviewer(s). You should upload this letter as a separate file labeled 'Response to Reviewers'.A marked-up copy of your manuscript that highlights changes made to the original version. You should upload this as a separate file labeled 'Revised Manuscript with Track Changes'.An unmarked version of your revised paper without tracked changes. You should upload this as a separate file labeled 'Manuscript'.

We look forward to receiving your revised manuscript.

Kind regards,

Leonard Ighodalo Uzairue, PhD

Academic Editor

PLOS ONE

Journal Requirements:

https://pdfs.semanticscholar.org/9e43/7517937937eb2d3e5b30178ad9c741781997.pdf

https://www.liebertpub.com/doi/10.1089/fpd.2021.0059

In your revision ensure you cite all your sources (including your own works), and quote or rephrase any duplicated text outside the methods section. Further consideration is dependent on these concerns being addressed.

“This work was partially supported by the BAS-USDA Program in Agriculture and Life Sciences [grant number BAS-USDA PALS LS-18] and Ministry of Science and Technology of Bangladesh [grant number SRG-221102].”

“This work was partially supported by the BAS-USDA Program in Agriculture and Life Sciences [grant number BAS-USDA PALS LS-18] and Ministry of Science and Technology of Bangladesh [grant number SRG-221102]. “

7. We note that Figure 1, 5 and 6 in your submission contain [map/satellite] images which may be copyrighted. All PLOS content is published under the Creative Commons Attribution License (CC BY 4.0), which means that the manuscript, images, and Supporting Information files will be freely available online, and any third party is permitted to access, download, copy, distribute, and use these materials in any way, even commercially, with proper attribution. For these reasons, we cannot publish previously copyrighted maps or satellite images created using proprietary data, such as Google software (Google Maps, Street View, and Earth). For more information, see our copyright guidelines: http://journals.plos.org/plosone/s/licenses-and-copyright.

1. You may seek permission from the original copyright holder of Figure 1, 5 and 6 to publish the content specifically under the CC BY 4.0 license.

8. PLOS ONE now requires that authors provide the original uncropped and unadjusted images underlying all blot or gel results reported in a submission’s figures or Supporting Information files. This policy and the journal’s other requirements for blot/gel reporting and figure preparation are described in detail at https://journals.plos.org/plosone/s/figures#loc-blot-and-gel-reporting-requirements and https://journals.plos.org/plosone/s/figures#loc-preparing-figures-from-image-files. When you submit your revised manuscript, please ensure that your figures adhere fully to these guidelines and provide the original underlying images for all blot or gel data reported in your submission. See the following link for instructions on providing the original image data: https://journals.plos.org/plosone/s/figures#loc-original-images-for-blots-and-gels.

Reviewers' comments:

Reviewer's Responses to Questions

**Comments to the Author**

1. Is the manuscript technically sound, and do the data support the conclusions?

Reviewer #1: Yes

Reviewer #2: Yes

2. Has the statistical analysis been performed appropriately and rigorously?

Reviewer #1: Yes

Reviewer #2: Yes

3. Have the authors made all data underlying the findings in their manuscript fully available?

Reviewer #1: Yes

Reviewer #2: Yes

4. Is the manuscript presented in an intelligible fashion and written in standard English?

Reviewer #1: Yes

Reviewer #2: Yes

Reviewer #1: General Comments

This manuscript addresses an important and timely topic, namely the prevalence of ESBL-producing and carbapenem-resistant E. coli and Salmonella spp. in poultry meat and live bird market (LBM) sewage in Bangladesh. The study is ambitious, covering all eight divisions of the country, and provides valuable epidemiological data with direct relevance for food safety and antimicrobial resistance (AMR) surveillance.

The paper is methodologically sound, but the manuscript is lengthy, data-heavy, and at times repetitive. The clarity of presentation can be improved by streamlining results, tightening the introduction and discussion, and emphasizing the key public health implications.

Specific comments are provided below.

Specific Comments

Abstract

Well done, it is recommended that only the main prevalence figures and key results be highlighted.

Introduction

Well contextualized, but partially repetitive (antibiotic misuse, poultry as reservoir). Condense to improve flow.

Explicitly state the knowledge gap: previous studies in Bangladesh were fragmented, this is the first comprehensive survey covering all divisions.

Methods

Sampling design is solid but justification for “5 samples per LBM” should be added.

Clarify whether clustering effects (design effect) were considered in sample size calculation.

Provide more detail on the positive and negative controls used in PCR assays.

Results

Results are highly detailed, with many percentages. Consider moving some data to supplementary materials.

Figures and maps are informative but crowded; merging or simplifying some would enhance readability.

Report confidence intervals along with prevalence estimates.

Discussion

The discussion is well written, but could perhaps be improved by better comparing more recent studies, if available, from South/South-East Asia and Africa.

Perhaps a clearer distinction could be made between explanations based on concrete evidence (e.g. the use of antibiotics) and more speculative interpretations.

Discuss in more detail the feasibility of the proposed interventions (antibiotic management, LBM sanitation services) in the context of Bangladesh.

Conclusion

The conclusions appear appropriate. Nevertheless, it would be advisable to include, among the practical recommendations, the strengthening of biosecurity measures throughout all stages of production and the implementation of stricter drug control. Such measures are consistent with practices adopted in other production systems, where antimicrobials are administered exclusively following a veterinary diagnosis and several active substances are restricted from use in livestock production.

Grammatical or stylistic corrections

Line 55- in the medical, veterinary and agriculture sectors.. correct in... medical, veterinary, and agricultural sectors.

Line 57-58 - ...through faces or manure... feces” (non “faces”).

Line 59-61 - ...and it is anticipated to climb by 67% by 2030 in rapidly polluting and developing countries around the world.... shorten the thought type.....and is projected to rise by 67% by 2030, particularly in rapidly developing countries.”

Line 64–67-“However, the chicken industry's fast expansion, combined with widespread and frequently uncontrolled antibiotic usage, has contributed to the spread of MDR bacteria across the food chain.” Better .. “The rapid expansion of the chicken industry, combined with uncontrolled antibiotic usage, has facilitated the spread of MDR bacteria through the food chain”.

Line 86-87 ...E. coli and Salmonella spp., that produces ESBLs, and demonstrates MDR, has implications...”

correct in... “that produce ESBLs and demonstrate MDR have implications...”

Line 93–94 - “...have demonstrated alarming levels of resistance...” better..high levels (eliminate alarming).

Line 109–110 - “...possible hotspot for ESBL-producing and carbapenem resistant bacteria to humans.” Better “hotspot for transmission ... to humans.”

Line 523 ....were MDR..” there is a double point

Line 526–535 -“The occurrence of high MDR and existance of pXDR among ESBL-producing and CR-E. coli and Salmonella spp. could be due to widespreed and unrestricted use of antibiotics during poultry production in Bangladesh as well as low levels of waste management system, which increases selective pressure for MDR bacteria, thus facilitating the emergence and dissemination of MDR foodborne bacteria in poultry production system and retail poultry products.” The sentence is too long. It should be split into at least two clauses.

Line 531–532 .....high MDR and existance of pXDR.... correct as “existence”.

Line 533....widespreed and unrestricted use... correct as “widespread”.

Line 555...ESBL-prodicing...” correct as .“producing”.

Line 606–608

“The drug regularity authority of Bangladesh should develop an implementable monitoring and evaluation system to tackle this issue. Besides, alternative therapies should be explored and assessed for treating and preventing multidrug resistant bacterial infections.” The sentence can be made more fluid: “The Bangladesh drug regulatory authority should implement effective monitoring systems, and alternative therapies should be explored to treat and prevent MDR infections.”.

drug regularity authority of Bangladesh...correct as “drug regulatory authority...”

Reviewer #2: Comments to authors:

The current study is interesting and clearly describes the findings with strong methedology; however, the authors are recommended to address the following comments to improve the overall quality of the manuscript:

Abstract

Clear and well-written

Introduction

1. Check for punctuation errors throughout the manuscript (double space, eg, line 60, comma eg. line 65)

2. Line 69. Among MDR strains, foodborne pathogens, especially Escherichia coli (E. coli) ____. The abbreviation should be introduced in the sentences above before using the short form

3. Line 82. Please cross-check that all scientific gene names are written correctly throughout the manuscript, eg. if CTX-M has to be in subscript format (blaCTX-M)

Method and material

4. Line 136-137. A cross-sectional live bird market (LBM) survey was carried out on 64 LBMs of 32 upazilas in 16 districts (second tier of administrative regions) of Bangladesh.

For better clarity, it is recommended to rewrite the sentence. Here are the suggested revisions: A cross-sectional survey of live bird markets (LBM) was conducted in 64 LBM locations across 32 upazilas in 16 districts (the second tier of administrative regions) of Bangladesh

5. Line 154-155. Prior to collection of samples, the informed consent was taken from the chicken’s seller.

I recommend rewriting this sentence, here is the suggested: Prior to the collection of samples, informed consent was taken from the chicken seller.

6. Line 212-213. The presence of ESBLs-encoding genes (blaCTX-M-1, and blaCTX-M-2) in ESBL-producing E. coli and Salmonella spp. isolates was determined by duplex PCR using specific oligonucleotide primers[35]

Question. 1. Why did you choose to screen only these ESBL genes? Why not include blaTEM and blaSHV as well?

Question. 2. Why did you choose to screen only one Cr gene?

7. line 230-231.………… commercially available imipenem (10 µg) and meropenem (10 µg) disks. Isolates that were resistant to at least one carbapenem drug were considered as CR isolates[34, 36], …..

These sentences contain wording errors and should be corrected: disk must be disks, …….on=one

8. Line 236-239. The PCR reaction mixtures of 25 µL total volume containing 12.5 µL of OneTaq® Quick-Load® 2X PCR Master Mix with standard buffer (New England, BioLabs Inc.), 8.5 µL water, 1 µL template DNA from each isolate, and 1.5 µL from each primer with a concentration of 15 pmol.

For better clarity, these sentences should be rewritten. You may consider the following suggested revisions.

The PCR reaction mixtures, with a total volume of 25 µL, contained 12.5 µL of OneTaq® Quick-Load® 2X PCR Master Mix with standard buffer (New England Biolabs Inc.), 8.5 µL of water, 1 µL of template DNA from each isolate, and 1.5 µL of each primer at a concentration of 15 pmol.

Result

9. Line 316-317. The carbapenemase gene, blaNDM-1, was detected in 5.2% (9/173) isolates of CR-E. coli and 33.6% (40/119) isolates of CR-Salmonella spp

Suggested correction The carbapenemase gene, blaNDM-1, was detected in 5.2% (9/173) of CR-E. coli isolates and 33.6% (40/119) isolates of CR-Salmonella spp.

10. Line 391-393. Isolates of ESBL-E. coli (40.6% and 21%) and Salmonella spp. (34.5% and 22.3%), as well as CR-E. coli (42.8% and 21.4%) and Salmonella spp. (42.9% and 25.2%) recovered from retail chicken meat, exhibited resistance to 9–12 and 5–8 antimicrobial agents, respectively

These sentences are unclear. What do the two percentages in one bracket represent (e.g., 40.6% and 21%; 34.5% and 22.3%)? Please rewrite them to avoid confusion for readers.

Discussion

11. Line 486. Interestingly, the blaCTX-M-2 gene was only detected in a single (2%) E. coli

Instead of using a word single … only one,

suggested correction. Interestingly, the blaCTX-M-2 gene was detected in only one (2%) E. coli

12. Line 516-518. Since different LBMs are anticipated to use different management techniques while slaughtering of chickens, and processing of chicken meat, there are varying hazards

Suggested corrections. Since different LBMs are anticipated to use different management techniques during the slaughter of chickens and processing of chicken meat, varying hazards are associated with the presence of CR-E. coli and Salmonella spp

13. Line 570-571 Additionally, Bangladesh should control the use of antibiotics in poultry in order to reduce the levels of resistance found in the current study.

Suggested correction. Additionally, Bangladesh should regulate the use of antibiotics in poultry to reduce the levels of resistance found in the current study.

14. In the discussion section, please include the public health implications of the high prevalence of MDR, ESBL-producing E. coli and Salmonella spp., as well as carbapenem-resistant E. coli and Salmonella spp. detected in sewage samples, and explain their potential role in disseminating antimicrobial resistance (AMR) into the environment

.

Reviewer #1: No

Reviewer #2: **Yes:** Mulu Lemlem DestaMulu Lemlem DestaMulu Lemlem DestaMulu Lemlem Desta

---

## [Author Response · Author response to Decision Letter 1]

22 Jan 2026

Response to reviewers:

Journal Requirements:

Response: We have carefully revised the manuscript to ensure full compliance with PLOS ONE style and formatting requirements, including file naming conventions.

Response: Though no specific permission was required for this fieldwork because the study was conducted in live bird markets that are open to the public, and the activities did not involve restricted areas, protected species, or regulated sampling. Verbal consent was taken from mayor of the city corporation or municipality of LBMs (Line 112-119).

https://pdfs.semanticscholar.org/9e43/7517937937eb2d3e5b30178ad9c741781997.pdf

https://www.liebertpub.com/doi/10.1089/fpd.2021.0059

Response: We have carefully reviewed the manuscript and revised the relevant sentences by rephrasing them to ensure originality of expression. We confirm that the data, analyses, and conclusions presented in this manuscript are entirely original. The manuscript has now been revised accordingly to minimize textual similarity.

In your revision ensure you cite all your sources (including your own works), and quote or rephrase any duplicated text outside the methods section. Further consideration is dependent on these concerns being addressed.

Response: We have carefully reviewed the entire manuscript to ensure that all sources, including our own previously published works, are appropriately cited wherever relevant. In addition, any instances of duplicated or closely similar text outside the methods section particularly in the introduction and discussion, have been fully rephrased. No verbatim text remains outside the methods section.

“This work was partially supported by the BAS-USDA Program in Agriculture and Life Sciences [grant number BAS-USDA PALS LS-18] and Ministry of Science and Technology of Bangladesh [grant number SRG-221102].”

Please provide an amended statement that declares *all* the funding or sources of support (whether external or internal to your organization) received during this study, as detailed online in our guide for authors at http://journals.plos.org/plosone/s/submit-now.

Please also include the statement “There was no additional external funding received for this study.” in your updated Funding Statement.

Response: The funding section has been updated (Line 565-569).

Response: We have added amended funding statement in the cover letter.

“This work was partially supported by the BAS-USDA Program in Agriculture and Life Sciences [grant number BAS-USDA PALS LS-18] and Ministry of Science and Technology of Bangladesh [grant number SRG-221102]. “

Response: This statement was already stated in the funding statement section of the original manuscript (Line 567).

Response: All relevant data is included in the manuscript and the Supporting Information file.

7. We note that Figure 1, 5 and 6 in your submission contain [map/satellite] images which may be copyrighted. All PLOS content is published under the Creative Commons Attribution License (CC BY 4.0), which means that the manuscript, images, and Supporting Information files will be freely available online, and any third party is permitted to access, download, copy, distribute, and use these materials in any way, even commercially, with proper attribution. For these reasons, we cannot publish previously copyrighted maps or satellite images created using proprietary data, such as Google software (Google Maps, Street View, and Earth). For more information, see our copyright guidelines: http://journals.plos.org/plosone/s/licenses-and-copyright.

Response: All maps now shown in Figures 1, 2a-d and S1 Fig are original, author-generated figures created using open-source QGIS software. No copyrighted, proprietary, or restricted sources (including Google Maps, Google Earth, or similar platforms) were used in their preparation. The shape files used for map generation were obtained from the website of the Humanitarian Data Exchange: (https://data.humdata.org/dataset/cod-ab-bgd).

1. You may seek permission from the original copyright holder of Figure 1, 5 and 6 to publish the content specifically under the CC BY 4.0 license.

Response: We did not use any copyright figures.

8. PLOS ONE now requires that authors provide the original uncropped and unadjusted images underlying all blot or gel results reported in a submission’s figures or Supporting Information files. This policy and the journal’s other requirements for blot/gel reporting and figure preparation are described in detail at https://journals.plos.org/plosone/s/figures#loc-blot-and-gel-reporting-requirements and https://journals.plos.org/plosone/s/figures#loc-preparing-figures-from-image-files. When you submit your revised manuscript, please ensure that your figures adhere fully to these guidelines and provide the original underlying images for all blot or gel data reported in your submission. See the following link for instructions on providing the original image data: https://journals.plos.org/plosone/s/figures#loc-original-images-for-blots-and-gels.

Response: We confirm that all figures containing blot or gel data in the manuscript have been reviewed to ensure full compliance with PLOS ONE’s blot and gel reporting and figure preparation guidelines. The original, uncropped, and unadjusted image files underlying all blot and gel results have now been provided as S1_raw_images, in accordance with the journal’s requirements. Adjustments applied to the images presented in the S1a-b Fig were limited to uniform brightness and contrast adjustments.

Response: Added in cover letter.

Reviewer #1: General Comments

This manuscript addresses an important and timely topic, namely the prevalence of ESBL-producing and carbapenem-resistant E. coli and Salmonella spp. in poultry meat and live bird market (LBM) sewage in Bangladesh. The study is ambitious, covering all eight divisions of the country, and provides valuable epidemiological data with direct relevance for food safety and antimicrobial resistance (AMR) surveillance. The paper is methodologically sound, but the manuscript is lengthy, data-heavy, and at times repetitive. The clarity of presentation can be improved by streamlining results, tightening the introduction and discussion, and emphasizing the key public health implications.

Response: Revised.

Specific comments are provided below.

Abstract

Well done, it is recommended that only the main prevalence figures and key results be highlighted.

Response: We have revised the relevant section to present the findings in a more focused and streamlined manner.

Introduction

Well contextualized, but partially repetitive (antibiotic misuse, poultry as reservoir). Condense to improve flow.

Explicitly state the knowledge gap: previous studies in Bangladesh were fragmented, this is the first comprehensive survey covering all divisions.

Response: The Introduction has been carefully revised to reduce redundancy.

Methods

Sampling design is solid but justification for “5 samples per LBM” should be added.

Clarify whether clustering effects (design effect) were considered in sample size calculation.

Provide more detail on the positive and negative controls used in PCR assays.

Response: A justification for collecting five samples per LBM has now been added to the Methods section. In addition, we have clarified the sample size calculation to address clustering effects (Line 131-136).

Detail on the positive and negative controls used in PCR assays have been provided (181-185; 210-212).

Results

Results are highly detailed, with many percentages. Consider moving some data to supplementary materials.

Figures and maps are informative but crowded; merging or simplifying some would enhance readability.

Report confidence intervals along with prevalence estimates.

Response: To improve clarity and readability, we have streamlined the Results section by reducing repetitive percentage reporting and consolidating parallel findings. Detailed resistance profiles and extended percentage data have now been moved to the supporting information section (S1 Fig, S1 Table and S2 Table). In addition, several maps have been simplified by merging overlapping panels and reducing visual complexity, while key findings are retained in the main figures.

Discussion

The discussion is well written, but could perhaps be improved by better comparing more recent studies, if available, from South/South-East Asia and Africa.

Perhaps a clearer distinction could be made between explanations based on concrete evidence (e.g. the use of antibiotics) and more speculative interpretations.

Discuss in more detail the feasibility of the proposed interventions (antibiotic management, LBM sanitation services) in the context of Bangladesh.

Response: We have revised the discussion section.

Conclusion

The conclusions appear appropriate. Nevertheless, it would be advisable to include, among the practical recommendations, the strengthening of biosecurity measures throughout all stages of production and the implementation of stricter drug control. Such measures are consistent with practices adopted in other production systems, where antimicrobials are administered exclusively following a veterinary diagnosis and several active substances are restricted from use in livestock production.

Response: We have revised the conclusion section to explicitly incorporate practical recommendations.

Grammatical or stylistic corrections

Line 55- in the medical, veterinary and agriculture sectors. correct in... medical, veterinary, and agricultural sectors.

Response: Corrected.

Line 57-58 - ...through faces or manure... feces” (non “faces”).

Response: Corrected.

Line 59-61 - ...and it is anticipated to climb by 67% by 2030 in rapidly polluting and developing countries around the world.... shorten the thought type.....and is projected to rise by 67% by 2030, particularly in rapidly developing countries.”

Response: Corrected.

Line 64–67-“However, the chicken industry's fast expansion, combined with widespread and frequently uncontrolled antibiotic usage, has contributed to the spread of MDR bacteria across the food chain.” Better. “The rapid expansion of the chicken industry, combined with uncontrolled antibiotic usage, has facilitated the spread of MDR bacteria through the food chain”.

Response: Corrected.

Line 86-87 ...E. coli and Salmonella spp., that produces ESBLs, and demonstrates MDR, has implications...”

correct in... “that produce ESBLs and demonstrate MDR have implications...”

Response: Corrected.

Line 93–94 - “...have demonstrated alarming levels of resistance...” better..high levels (eliminate alarming).

Response: Corrected.

Line 109–110 - “...possible hotspot for ESBL-producing and carbapenem resistant bacteria to humans.” Better “hotspot for transmission ... to humans.”

Response: Corrected.

Line 523 ....were MDR..” there is a double point

Response: Corrected.

Line 526–535 -“The occurrence

---

## [Decision Letter · Decision Letter 1]

30 Mar 2026

Distribution of ESBL-producing and carbapenem-resistant E. coli and Salmonella spp. in retail chicken meat and live bird market sewage in Bangladesh

PONE-D-25-41667R1

Dear Dr. Islam,

We’re pleased to inform you that your manuscript has been judged scientifically suitable for publication and will be formally accepted for publication once it meets all outstanding technical requirements.

Kind regards,

Leonard Ighodalo Uzairue, PhD

Academic Editor

PLOS One

Additional Editor Comments (optional):

Reviewers' comments:

Reviewer's Responses to Questions

**Comments to the Author**

Reviewer #1: (No Response)

2. Is the manuscript technically sound, and do the data support the conclusions?

Reviewer #1: (No Response)

3. Has the statistical analysis been performed appropriately and rigorously?

Reviewer #1: (No Response)

4. Have the authors made all data underlying the findings in their manuscript fully available?

Reviewer #1: (No Response)

5. Is the manuscript presented in an intelligible fashion and written in standard English?

Reviewer #1: (No Response)

Reviewer #1: Distribution of ESBL-producing and carbapenem-resistant E. coli and Salmonella spp. in retail chicken meat and live bird market sewage in Bangladesh

General Comment

The research examines numerous live bird markets and their respective wastewaters in Bangladesh. The authors investigated the presence of antibiotic-resistant bacteria, specifically E. coli and Salmonella spp, in both retail chicken meat and sewage effluent. The findings reveal a high prevalence of these species, particularly those exhibiting multidrug resistance (MDR), including resistance to last-resort antibiotics such as carbapenems.

As the authors rightly highlight, these results underscore the urgent need for action by health authorities to limit antibiotic use in livestock and to improve hygiene and biosecurity measures within Bangladesh's live poultry markets.

The authors employed both classical and molecular methods, which successfully addressed the objectives outlined in the introduction. The results are interpreted and discussed correctly.

Following the revisions made to the manuscript, the paper is now suitable for publication without further changes.

.

Reviewer #1: **Yes:** Dr. Guido GrilliDr. Guido GrilliDr. Guido GrilliDr. Guido Grilli

---

## [Editor Report · Acceptance letter]

PONE-D-25-41667R1

PLOS One

Dear Dr. ISLAM,

I'm pleased to inform you that your manuscript has been deemed suitable for publication in PLOS One. Congratulations! Your manuscript is now being handed over to our production team.

Kind regards,

on behalf of

Dr. Leonard Ighodalo Uzairue

Academic Editor

PLOS One